# The Influence of Cultural Factors on Choosing Low-Emission Passenger Cars

**Ioana Ancuta Iancu [1], Patrick Hendrick [1], Dan Doru Micu [2], Denisa Stet [2], Levente Czumbil [2] and Stefan Dragos Cirstea [3],***

[1] Aero-Thermo-Mechanics, Universite Libre de Bruxelles, 1050 Bruxelles, Belgium; ioana-ancuta.iancu@ulb.be (I.A.I.); patrick.hendrick@ulb.be (P.H.)

[2] Electrotechnics and Measurement, Technical University of Cluj-Napoca, 400114 Cluj-Napoca, Romania; dan.micu@ethm.utcluj.ro (D.D.M.); denisa.stet@ethm.utcluj.ro (D.S.); levente.czumbil@ethm.utcluj.ro (L.C.)

[3] Power Systems and Management, Technical University of Cluj-Napoca, 400114 Cluj-Napoca, Romania

* Correspondence: stefan.cirstea@enm.utcluj.ro; Tel.: +40-74-530-1093

**Abstract:** The decrease in greenhouse gas emissions by passenger cars is one of the key factors for climate protection measures. Besides EU strategies for low-emission mobility, policy makers must consider the behavioural factors of buyers. This study aims to cover this gap by investigating the relation between the national cultural dimensions (Hofstede model) and car adoption by fuel type in EU countries. This could help car sellers to find better solutions for advertising cars with medium and low greenhouse gas emissions. To find better ways to increase the usage of medium- and low-emission cars using targeted advertising, correlations and a multiple regression analysis were used. The results show that the consumer preference for one type of fuel is correlated with at least one of Hofstede's six cultural dimensions: the power distance index; individualism versus collectivism; masculinity versus femininity; the uncertainty avoidance index; long-term orientation versus short-term normative orientation; indulgence versus restraint. The major conclusion of the study underlines that, with increases in the individualism versus collectivism and indulgence versus restraint scores, the usage of low- and medium-emission cars also increases, and with the increase in the power distance and uncertainty avoidance index, the usage of low- and medium emission cars decreases. At the same time, the driving preference for low- and medium-emission vehicles decreases with the tendency towards collectivism and restraint of EU countries.

**Keywords:** battery electric vehicles; plug-in hybrid electric vehicles; hybrid electric vehicles; $CO_2$ emissions; Hofstede; advertising

## 1. Introduction

The climate and environment changes has become one of the most important topics in recent years. The release of carbon dioxide ($CO_2$) and other greenhouse gas (GHG) emissions due to human activities has led to global warming [1] and, therefore, to climate change. Factors such as the increases in urbanization, population, wealth, energy consumption, and agriculture activities have resulted in environmental change [2]. As a result, the European Green Deal for the European Union (EU) emerged. One of the aims of this commitment is to transform the EU into a region where there are no net GHG emissions by 2050 [3].

Worldwide, the transport sector was responsible for 16.2% of the GHG emissions [4], and it was the main pollutant in Europe [5], responsible for 22.3% of the total GHG emissions in 2020. Most of the GHGs come from passenger cars (44%) [6]. Therefore, the reduction in GHGs for passenger cars is one of the key factors for climate protection measures [7]. Besides EU strategies for low-emission mobility, policy makers must consider the behavioural factors of buyers. Buying behaviour is influenced by cultural, personal, social, and psychological factors [8]. A large body of literature focuses on three of them

(personal, social, and psychological factors), and only a few studies focus on the cultural characteristics of car buyers by fuel type. Furthermore, the studied literature shows that considering only the incentives when promoting a certain type of fuel will influence buyer behaviour only for a short period of time. We do not argue the relevance of price, incentives, taxes, and other already studied factors that affect the decision to buy a car with a certain engine.

The objective of our study is to find out how specific traits of national culture, as described by Hofstede's culture dimensions, influence the usage of low-emission passenger cars. To complete the objective, after reviewing the literature, the first step was to find out whether there is a correlation between the cultural dimensions and fuel choice when driving a passenger car. Using multiple regression analysis, we found which cultural traits can best predict the choice of a passenger car.

To the best of our knowledge, there is no similar study that shows how to improve the marketing strategies of low-emission passenger cars using culture as a determinant. This study aims to complete the literature by analysing the relations between the national cultural dimensions and car adoption by fuel type in EU countries. By taking into consideration cultural factors, this study could help policy makers, scientists, and marketing specialists find better solutions for promoting the sales of cars with low $CO_2$ emissions by properly addressing them to each country.

To accomplish the research objective, the paper begins with a concise overview of the research topic, and it then reveals an extensive literature review in Section 2, which explores carbon dioxide emissions from passenger cars and the cultural factors that impact the purchasing decisions for these vehicles. Section 3 provides details on the research methodology, while Section 4 presents the findings, results, and a discussion of the study. Finally, the paper concludes with a summary of the key insights from the research in the last section.

## 2. Literature Review

### 2.1. Carbon Dioxide Emissions of Passenger Cars by Fuel Type

Inside the EU, in the last years, the total GHG emissions have dropped, but the transport sector has not followed the same trend. Moreover, in 2019, GHG emissions increased by 0.8% (shipping was not included) [9]. The emissions of GHGs in the transport industry consist mainly of NOx, CO, and NMVOCs, the bigger share being $CO_2$ [10].

In the transport sector, passenger cars and light-duty vehicles are the main pollutants, and together they are responsible for 70% of the total GHG emissions in the EU [11]. Due to this matter, the EU is obliged to find new ways to address and encourage the acquisition of less polluting passenger cars.

On EU roads, in 2021, the passenger cars by fuel type were as follows: 52.9% petrol; 42.3% diesel; 3.4% alternative fuels; 0.8% hybrid; 0.4% electric cars (BEVs) [12]. The most used passenger cars are those with internal combustion engines, which are the ones that emit the most GHGs [13]. Figure 1, which represents the average $CO_2$ emissions from different fuel types, shows that BEVs have zero emissions, and that the cars that use E85, LPG, diesel, and petrol as fuel pollute the most.

The literature shows that cars that use bioethanol (E85) have the highest $CO_2$ emissions (Figure 1), but they have the lowest total greenhouse effect when sugar cane-based fuel is used. The reason behind this contradiction dwells in the uptake of $CO_2$ from the air during the growth of sugar cane [14]. In the EU, the countries with the highest rates of $CO_2$ emissions from new cars that come from E85 are Germany (292 g$CO_2$/km) and France (185.5 g$CO_2$/km) [15].

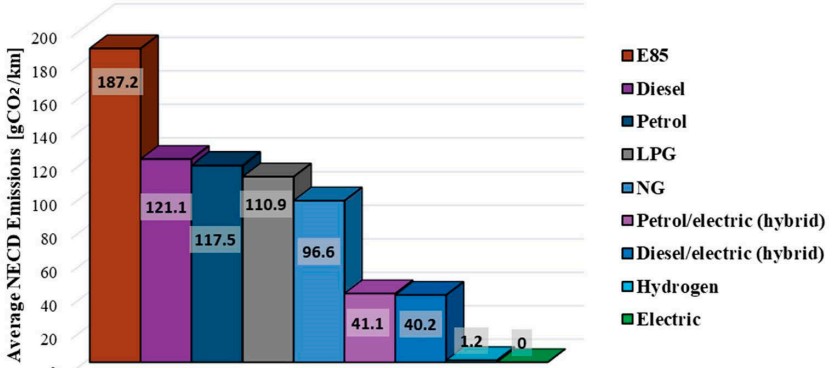

**Figure 1.** Average NECD emissions (gCO$_2$/km) from new passenger cars in EU countries (2020). Source: [14].

Due to the high rate of GHG emissions from diesel cars, many cities in the EU are considering banning them in the central areas [16]. Diesel cars are mostly used in Lithuania and Latvia (69.2% and 63.2%, respectively, from the total fleets of passenger cars), and they are used the least in Greece (8.1%) [17]. Petrol is the most used fuel in the EU (52.9%), ranging between 91.1% of the cars in Greece and 26.1% of the cars in Lithuania [18].The popularity of petrol and diesel cars resides in the range of vehicles being designed for long-distance trips [18]. However, a 2019 study made in Portugal shows a decrease by up to −27% in emissions when cleaner vehicles are used that comply with the post-Euro 6 standards [19].

LPG is a colourless gas that is derived from petroleum, and it is most often used when converting the existing technologies of passenger cars to run on cheaper and lower GHG emissions [20]. From the 9,787,916 passenger cars registered in the EU in 2020, 151,999 used LPG as fuel, which emits, on average, 110.9 NECD (gCO$_2$/km). In this case, the differences between the CO$_2$ emissions in EU countries are small, ranging between 121.4$_2$ /km in Belgium and 102.3 gCO$_2$/km in Romania [14]. Most of the registered cars that use LPG are in Italy (93,339 (6.3% of the cars)), with fewer in Luxemburg (only five cars) [14,17]. Compressed natural gas (CNG) is used by 0.5% of the passenger car fleet in the EU, being most common in Italy (2.4%) [17]. In 2020, the average emissions of gCO$_{22}$/km from new registered cars ranged between 127.4 (Poland) and 81 (Croatia) [9]. CNG is described as one of the promising low-emission alternatives for the short- and mid-term decarbonization of road transport in the EU [21], but studies show that there are no benefits in terms of GNG emissions [22].

The below figure (Figure 2) shows the distribution of high-emission passenger vehicles in EU-27. The most passenger cars that use fuels with high polluting rates (petrol, diesel, LPG, GNG) are in Latvia, Croatia, Slovenia, and Estonia, while Sweden, the Netherlands, and Lithuania have less.

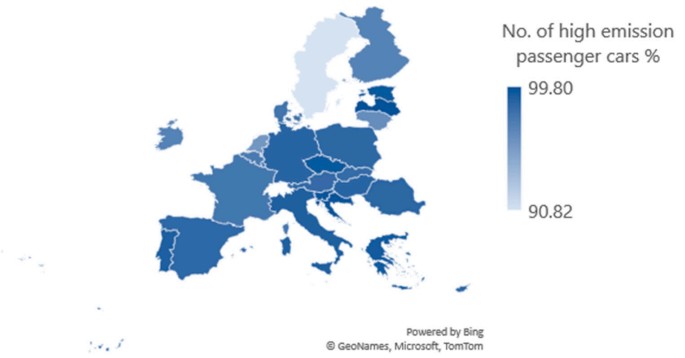

**Figure 2.** Percentages of passenger cars in EU-27 (except Bulgaria) with high CO$_2$ emissions in 2019. Source: all the data from the ACEA [14], except for the following: Austria—all passenger cars [23]; Croatia—BEVs [24]; Cyprus, Estonia, France, Latvia, and Malta—all passenger cars [24]; Denmark—LPG [25].

The differences between the average emissions from new petrol–electric and diesel–electric cars are small (41.1 and 40.2 gCO$_2$/km, respectively), being mostly preferred by Germans in the EU [14]. Hybrid engine cars (HEVs) are a combination of the combustion engine and an electric motor, and they are an intermediate solution between ICVs and BEVs [18]. These cars are mostly used in Sweden, the Netherlands, and Ireland (2.4% of the total passenger car fleet), and they are less used in Croatia and Romania (0.2% of the total passenger car fleet) [17]. Another solution is plug-in electric cars (PHEVs), which are cars can be directly charged from the power grid and can be driven at 20–50 km using only electricity [26]. It is safe to say that, on small trips, the CO$_2$ emissions of PHEVs are zero.

The available data (Figure 3) show higher rates of registered HEV and PHEV passenger cars in Sweden, the Netherlands, Finland, Ireland, and Belgium, and fewer in Poland, Romania, Croatia, and Latvia.

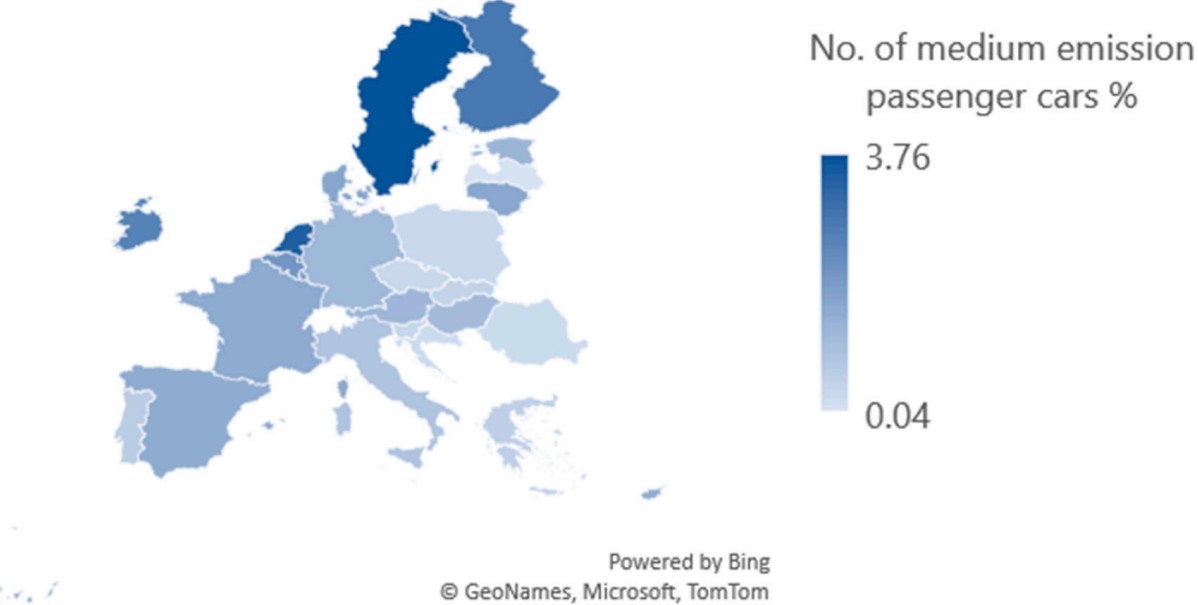

**Figure 3.** Percentages of passenger cars in EU-27 (except Bulgaria) with medium CO$_2$ emissions in 2019. Source: all the data from ACEA [14], except for the following: Austria—all passenger cars [23]; Croatia—BEVs [24]; Cyprus, Estonia, France, Latvia, and Malta—all passenger cars [24]; Denmark—LPG [25].

A 2021 study shows that only electric and hydrogen fuel can help in achieving the goals of the Paris Agreement [27]. As the above figure illustrates (Figure 4), hydrogen used as a fuel for passenger cars emits only 1.2 g CO$_{22}$/km, and BEVs have 0 gCO$_{22}$/km emissions. Hydrogen cars use H$_2$ to generate electricity, the main advantage being that they are easily refuelled at filling stations in 3–5 m and they have a good driving range [26]. Many studies show that using electricity to power BEVs has the lowest climate impact. The Netherlands has the highest rate of BEVs (1.2%), followed by Austria, Denmark, and Sweden (0.6%) [17]. A study from 2018 indicates that GHG emissions are reduced by 50–60%, on average, when using BEVs compared with internal combustion engines [28]. The collected data show that cars that use hydrogen and electricity as fuel are more common in the Netherlands, Sweden, Denmark, and Luxembourg.

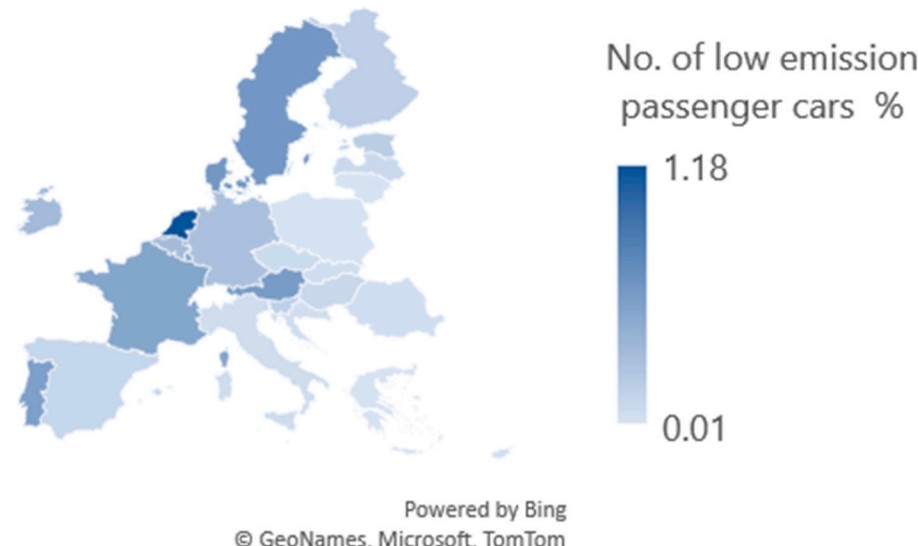

**Figure 4.** Percentages of passenger cars in EU-27 with low $CO_2$ emissions in 2019. Source: all the data from ACEA [14], except for the following: Austria—all passenger cars [23]; Croatia—BEVs [24]; Cyprus, Estonia, France, Latvia, and Malta—all passenger cars [24]; Denmark—LPG [25].

To decrease the $CO_2$ emissions, which are the main component of GHGs, the policy makers in the EU use almost 700 measures to address road transport emissions [11], and to decrease the demand for polluting cars and promote the use and production of more energy-efficient vehicles [16,29,30]. They use "push" or "pull" strategies, [31,32]. The first type of strategy addresses the car manufacturers and fuel suppliers, and the second (pull strategies) applies to the demand side [11]. EU and national policies related to taxes (on fuel, vehicles, emissions) can decrease the demand for polluting cars and aim to promote the production of more fuel-efficient vehicles. Besides EU and national strategies for low-emission mobility, policy makers must consider the behavioural factors of buyers. Through economic incentives, buyers are motivated only for the period during which they have benefits; afterwards, they will return to their old buying habits [33–35]. Another study demonstrates that besides financial help, it is necessary to have a certain level of self-sustainability [36].

### 2.2. Cultural Factors Influencing Passenger-Car-Buying Behaviour

Buying behaviour is influenced by cultural, social, personal, and psychological factors [37]. A large body of literature focuses on the personal, social, and psychological factors that influence passenger-car-buying behaviour when choosing the fuel type of the car, but only a few focus on the cultural characteristics of buyers.

Reviewing the literature, we find many diverse definitions of culture. In 1952, Kroeber and Kluckhohn listed 160 definitions [38]. Some authors explain culture through empirical studies, while others use more generic formulations [39]. Kotler and Armstrong explain culture as a cumulus of fundamental values, perceptions, wishes, and learned behaviours, which are different from one society to another [8]. Moreover, they consider culture as the most profound cause that influences consumer behaviour [37].

One of the most cited authors, Schwartz, sees culture as a "complex of meanings, beliefs, practices, symbols, norms, and values prevalent among people in a society" [40]. The model developed by the author introduces seven dimensions that can predict consumer behaviour: intellectual autonomy; affective autonomy; embedded cultures; cultural egalitarianism; cultural hierarchy; harmony; mastery [40].

Hofstede defines culture as a collective "programming of the mind" phenomenon [41,42]. He explains that culture represents a set of elements, such as beliefs, attitudes, collective activities, role models, and the language common to a particular group [43]. By repeated

empirical research, Hofstede created one of the most comprehensive models with which we can characterize national cultures. His model provides a scale from 0 to 100 for all the dimensions, by country [42].

Hofstede's six cultural dimensions are as follows: the power distance index (PDI); individualism versus collectivism (IDV); masculinity versus femininity (MAS); the uncertainty avoidance index (UAI); long-term orientation versus short-term normative orientation (LTO); indulgence versus restraint (IND).

The model has been used in various sectors and for various perspectives; for example, for the prediction of proenvironmental behaviour in hospitality and tourism [44], changing the organisational culture to increase innovation and productivity [45], understanding the perceived risks related to self-driving cars [46], etc.

The PDI reflects the acceptance of the power distribution in a society. When a country has a score close to 100, the less powerful members will more easily accept the hierarchy and inequalities [47,48]. In countries with high PDIs (Figure 5), luxury articles, fashion items [49], and expensive cars [50] are used to make one's status clear.

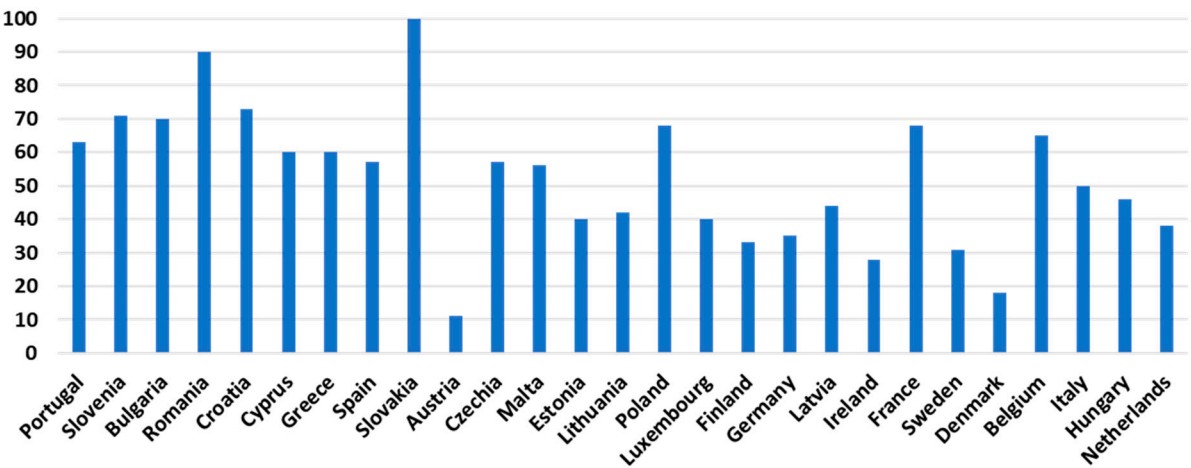

**Figure 5.** EU-27 countries' power distance indexes (PDIs) according to Hofstede's culture dimension model [42].

In individualist cultures (Figure 6), people look after themselves and immediate family, and they are striving for identity [47]. Furthermore, they use explicit communication, contrary to collectivist cultures. The countries that score low in this dimension consider that to sell something, first you must create trust [49]. In collective cultures, people are more inclined to develop a pro-environmental attitude, being ready to pay more for the wellbeing of all society [51]. A study conducted in Germany, Mexico, and Spain shows that a higher level of collectivism develops stronger eco-friendly behaviours, and stronger intentions to adopt renewable energy technologies [52]. The preferred advertisements in collectivist countries are focussed on the idea of team, collaboration, and the victory of the community [53].

If in a masculine society (Figure 7), the main drivers are achievement and success, in a feminine society, the values are caring for others and a good life quality [49]. The same study shows that dimension plays an important role: big in the masculine dominated societies and small in the feminine ones. The advertising in countries dominated by masculine values is focused on success (by showing luxury brands) [49], competitiveness, dreams, expectations, and nonfictional elements [54].

The UAI refers to the ways in which the individuals in a society relate to uncertainty and ambiguity [54]. In cultures with high scores for this dimension (Figure 8), the individuals are more resistant to accepting new technologies and innovation [55], the conflicts are threatening, individuals have an aggressive driving style [54], and advertisements are structured and serious with a great deal of technical information [56].

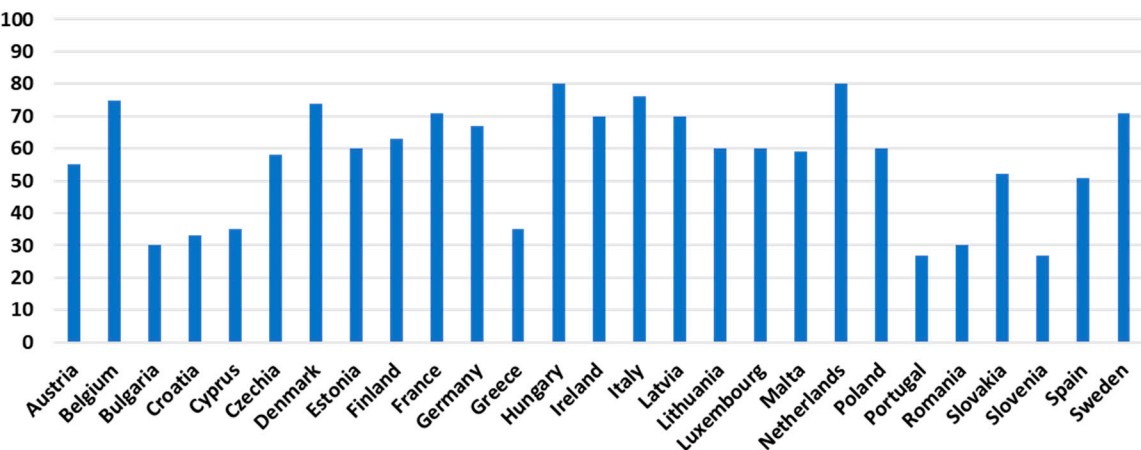

**Figure 6.** EU-27 countries' individualism indexes (IDVs) according to Hofstede's culture dimension model [42].

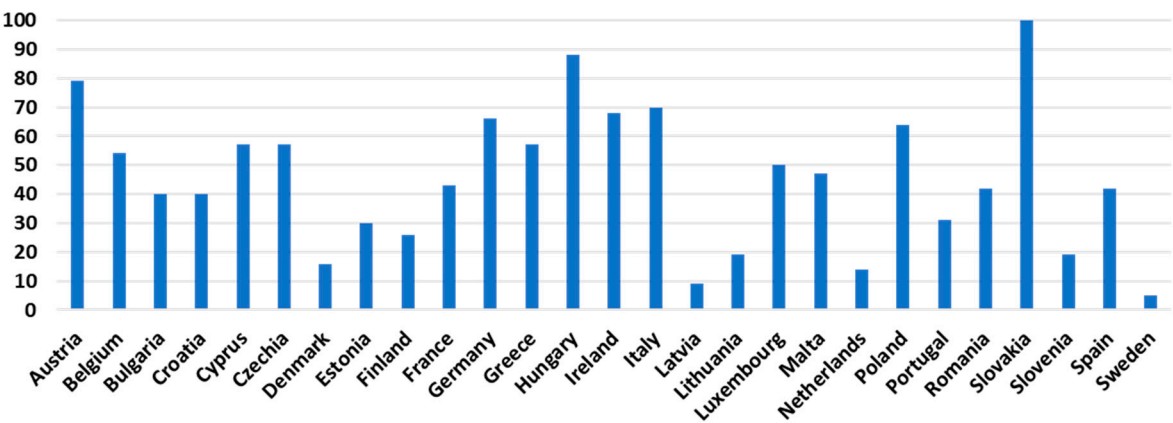

**Figure 7.** EU-27 countries' masculinity indexes (MAS) according to Hofstede's culture dimension model [42].

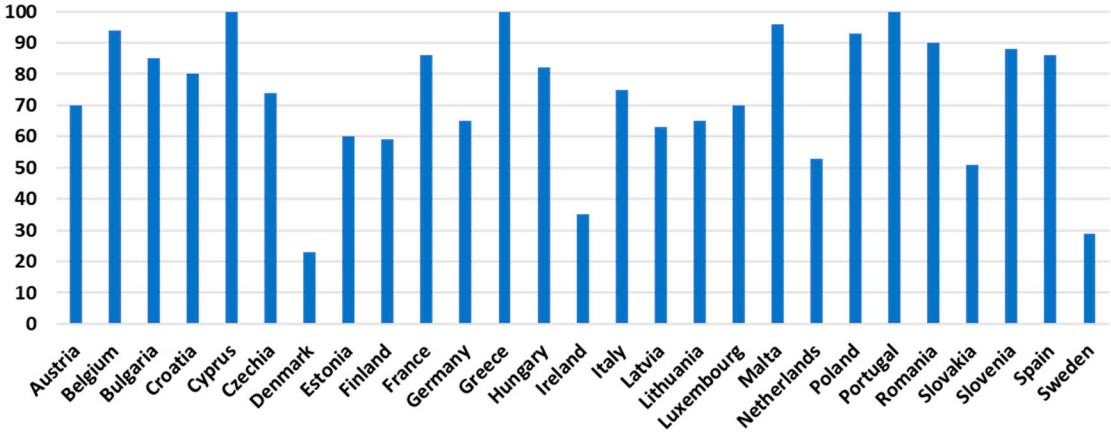

**Figure 8.** EU-27 countries' uncertainty avoidance indexes (UAIs) according to Hofstede's culture dimension model [42].

When a society is driven by elements such as perseverance [49], pragmatism, and a focus on the future [48], it scores high in the LTO dimension (Figure 9). Moreover, LTO corelates with pro-environmental behaviours [57].

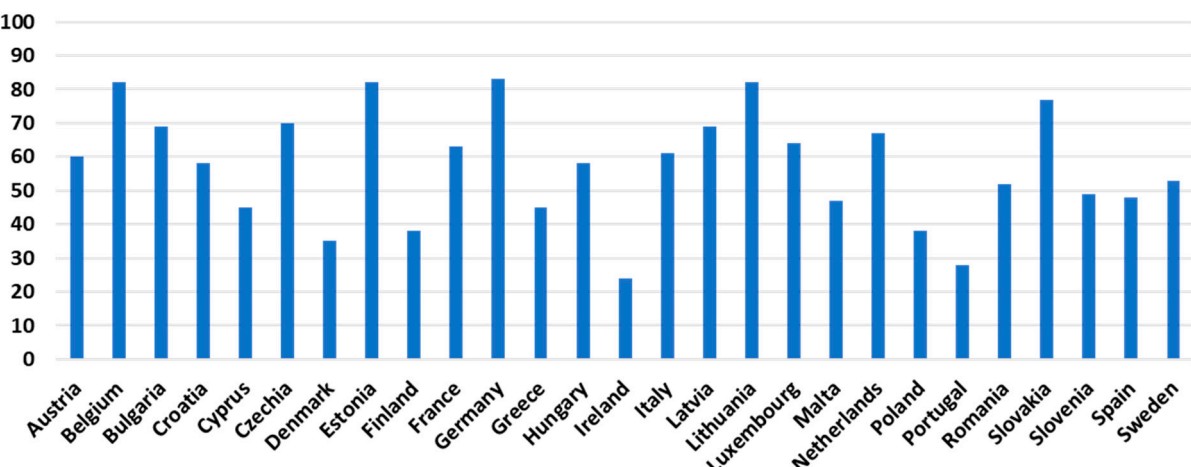

**Figure 9.** EU-27 countries' long-term orientation indexes (LTO) according to Hofstede's culture dimension model [42].

The last dimension, IND, was introduced in 2010, and it describes the inclination of a society to enjoy life in opposition to the ones that suppress gratification [58]. Usually, people living in restraint societies with low IND indexes (Figure 10) tend to be more cynical and pessimist [59]. This is the least studied dimension in the literature.

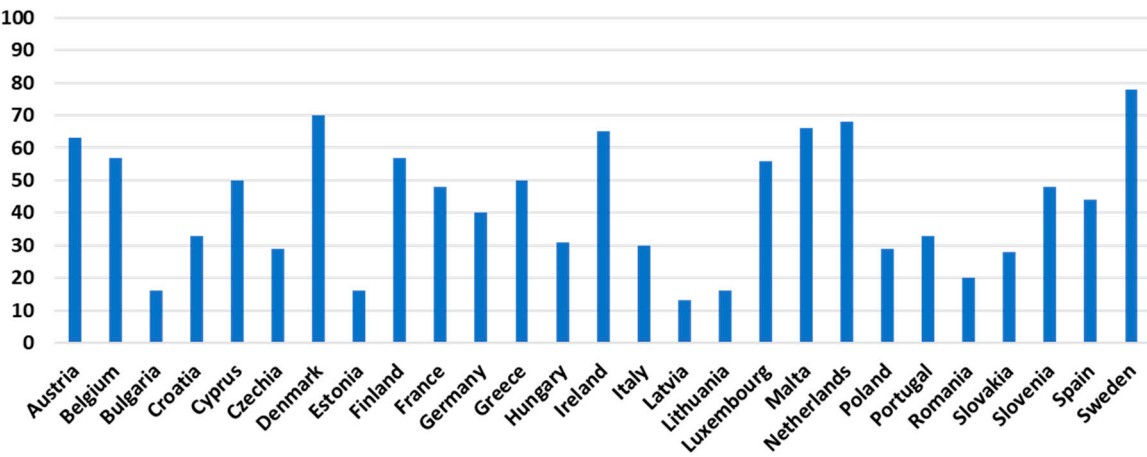

**Figure 10.** EU-27 countries' indulgence indexes (INDs) according to Hofstede's culture dimension model [42].

The Hofstede model is debated by some authors because heterogenous groups can live within a state, but we must consider that sharing the same education system, healthcare system, legal system, and institutions makes them share a common goal over a long period [43], and if it is required, Hofstede's model could be applied to smaller communities as well. Many articles demonstrate the relevance of this model for cross-cultural studies in marketing, psychology, sociology, or management [60], as it is the best way to measure national cultures [61]. The arguments that support the choice of using Hofstede's cultural dimensions in the present paper are described next. Hofstede's cultural dimensions have been used for a long time (since 1970) and have an extensive empirical base from all over the world. His research was conducted in the 1970s, and it has been expanded and validated over time, with his dimensions being used in numerous studies across different countries and cultures. Hofstede's dimensions are recognized and used in the business world so that companies can tailor their marketing strategies to different cultures [62].

McLeay et al. recommend that future studies on eco-friendly consumption should consider cultural contexts [63]. Barbarossa et al. compared the EV adoption intentions

among consumers in Denmark, Belgium, and Italy using Hofstede's cultural dimensions to explain cross-national differences [33]. According to the results, public policy and social marketing campaigns should give greater consideration to the impact of cultural values when promoting environmentally sustainable technologies. It is important to note that promotional efforts should vary based on the specific cultural elements and the level of innovation of the products being promoted [63].

## 3. Research Methodology

The purpose of this study is to establish the influence of national culture, as described by Hofstede's cultural dimensions, on the selection of fuel for automobile usage by following a series of methodological steps to arrive at a conclusive answer.

First, a literature review and a collection of data from different sources were made. Afterwards, an assessment of the passenger cars by fuel type in every country of the EU was conducted. The percentages of passenger cars by fuel type were taken into consideration, and not the number of passenger cars in the EU, in 2019. The authors consider that new car purchases during the COVID-19 pandemic years were more influenced by the availability of cars and their prices on the market (the component crisis) than the cultural dimensions of the buyers. The share of each chosen fuel type when driving a car is more important than the effective number because it can show the preferred passenger car in a country. In the study, all the data come from five official sources: the ACEA [14], except for the following: Austria—all passenger cars [23]; Croatia—BEVs [24]; Cyprus, Estonia, France, Latvia, and Malta—all passenger cars [24]; Denmark—LPG [25]. From the EU countries, Bulgaria is the only one that does not have any official data on the number of diesel, petrol, and hybrid electric passenger cars. Bulgaria was not excluded from the study, but the absent information was noted as missing data.

The second step was to collect data on the six dimensions of Hofstede's model from the homepage Geert Hofstede [63]. The Hofstede model of cultural dimensions was last updated in 2010 within [42]. This edition includes an expanded analysis of cultural differences and updates to the original cultural dimensions based on data collected from additional countries and regions. Each dimension, for all the studied countries, has a score between 1 and 100. Cyprus was the only country with no characterization of its cultural dimensions. However, at least one study assumes that due to the similarities with Greek culture, the same score can be applied [33].

Next, the data were introduced in SPSS 28 software, and a descriptive analysis of the elements was made. To observe the existence of links between the preferred fuel in passenger cars and the cultural dimensions, we used the normal distribution of the data, using Shapiro–Wilk tests, histograms, and plots. Further, Pearson's correlation for the normally distributed data ($p > 0.05$) and Spearman's rho for the non-normally distributed data ($p < 0.05$) were used.

The results show the strength of the correlations between two variables and can range between $-1$, which is a perfect negative correlation, and $+1$, which is a perfect positive correlation. When the coefficient is close to 0, the results show no relation between the variables. It is mandatory to consider the value of the significance (sig 2-tailed), which must be less than 0.05 [64].

For answering the research question, three groups of passenger cars by $CO_2$ emissions were considered:

- Passenger cars with low $CO_2$ emissions, which were calculated by adding the percentage of BEVs and $H_{22}$;
- For medium $CO_2$ emissions, we considered data on PHEVs and HEVs;
- For high-$CO_{22}$-emission vehicles, the usage of petrol, diesel, LPG, and NG as fuel (Figure 1) from all the studied countries.

To explain the usage of high-, medium-, and low-emission passenger cars by the cultural dimensions, multiple regression analysis was applied. The first step was to find out whether there is a correlation between passenger vehicles by $CO_2$ emissions and the

cultural dimensions. Using Shapiro–Wilk tests, we found which correlation tests are the best fit. Pearson's correlation for normal distribution data or Spearman tests for non-normal distribution data were used.

Next, in the multiple regression analysis, the cultural dimensions are considered as independent variables, and one of three groups of vehicles by $CO_2$ emissions with which a correlation was found is the dependent variable. For the multiple regression, the equation used was as follows:

$$Y = \beta_0 + \beta_1 \cdot X_1 + \beta_2 \cdot X_2 + \cdots + \beta_k \cdot X_k$$

where $Y$ is a dependent variable (percentage of high-, medium-, or low-emission passenger cars); from $X_1$ to $X_k$ are the independent variables (the cultural dimensions), considering a total of $k$ independent variables; from $\beta_1$ to $\beta_k$ are the regression coefficients; $\beta_0$ is the intercept value (regression constant).

To identify the most significant cultural dimensions that explain the usage of high-, medium-, or low-emission passenger cars, the backward elimination method was applied. This is a widely used method that starts from a multilinear regression model that contains all the independent variables that are of interest and eliminates, one by one, the independent variables that are not significant in predicting the dependent variable until the highest accuracy regression model is identified.

## 4. Results and Discussion

The descriptive analysis of the fuels used for passenger cars in the EU gives a picture of the substantial differences between the consumer preferences in each country in 2019 (Table 1). The differences between the lowest and highest rates of registered cars in different countries are given by the minimum and maximum values. It can be observed that there are countries with low percentages of passenger cars that use LPG as fuel (Ireland or Sweden), CNG (Malta, Cyprus, Croatia, Latvia), and HEVs (Latvia, Croatia, Romania), and other countries with high rates of these types of cars. Even if studies show the importance of $H_{22}$ in the reduction in the $CO_2$ for passenger cars, due to barriers such as price, infrastructure, and distribution [65], this fuel is less used. Petrol- and diesel-fuelled engines are the most common in the EU, with means of 54.23% and 42.07%, respectively.

**Table 1.** Descriptive statistics of variables.

|  | N | Minimum | Maximum | Mean | Std. Deviation |
|---|---|---|---|---|---|
| **HIGH_EMISS** | 26 | 91.46 | 99.8 | 98.009 | 1.7106 |
| Petrol | 26 | 26.1 | 91.1 | 54.231217 | 15.8454203 |
| Diesel | 26 | 8.1 | 69.2 | 42.068731 | 15.5571305 |
| LPG | 27 | 0 | 13.6601 | 1.827963 | 3.1714756 |
| CNG | 27 | 0 | 2.4402 | 0.216387 | 0.4929744 |
| **MED_EMISS** | 26 | 0.04 | 3.76 | 1.279 | 0.99596 |
| HEV | 25 | 0 | 2.4 | 1.02 | 0.7147261 |
| PHEV | 27 | 0.0098 | 1.3627 | 0.233959 | 0.3619076 |
| **LOW_EMISS** | 27 | 0.01 | 1.18 | 0.2641 | 0.28539 |
| $H_2$ | 27 | 0 | 0.0038 | 0.000394 | 0.0008715 |
| BEV | 27 | 0.0081 | 1.1748 | 0.264762 | 0.2796641 |

Besides the economic-related factors, the cultural dimensions of the European countries may explain some of these differences.

To find out whether there is a link between the chosen fuel type and the national cultural dimensions, first, the data were tested for normal distribution (Table 2). Using Shapiro–Wilk tests, the results show that data for petrol, diesel, HEVs, low emissions, medium emissions, high emissions, the PDI, MAS, the UAI, LTO, and IND are normally distributed (sig. > 0.05). The rest of the variables (LPG, CNG, PHEVs, $H_2$, BEVs, and IDV) are not normally distributed (sig. < 0.05)

**Table 2.** Normal distribution of variables.

| | Shapiro–Wilk Test | | | | | | |
|---|---|---|---|---|---|---|---|
| | **Statistic** | **df** | **Sig.** | | **Statistic** | **df** | **Sig.** |
| **Petrol** | 0.979 | 25 | 0.856 | **BEV** | 0.809 | 25 | <0.001 |
| **Diesel** | 0.977 | 25 | 0.81 | **PDI** | 0.983 | 25 | 0.93 |
| **LPG** | 0.659 | 25 | <0.001 | **IDV** | 0.906 | 25 | 0.025 |
| **CNG** | 0.436 | 25 | <0.001 | **MAS** | 0.974 | 25 | 0.759 |
| **HEV** | 0.923 | 25 | 0.06 | **UAI** | 0.936 | 25 | 0.122 |
| **PHEV** | 0.673 | 25 | <0.001 | **LTO** | 0.962 | 25 | 0.453 |
| **H$_2$** | 0.53 | 25 | <0.001 | **IND** | 0.953 | 25 | 0.286 |

*4.1. High-Emission Passenger Cars*

To observe the link between the variables, a Pearson correlation test was applied for the normally distributed data. The next table (Table 3) shows that the percentage of petrol passenger cars in EU-27 countries has a moderate positive correlation with the IND cultural dimension (r = 0.4190, sig. = 0.03). This indicates that the inclination to buy petrol-fuelled cars is linked to the tendency of people living inside national borders to enjoy life, not being influenced by social norms.

**Table 3.** Correlations between cultural dimensions and usage of high-emission passenger cars. *, **—Automated highlight of SPSS software to underline greatest values.

| | | PDI | IDV | MAS | UAI | LTO | IND |
|---|---|---|---|---|---|---|---|
| **Petrol** | Pearson Correlation | −0.119 | 0.122 | 0.096 | 0.01 | −0.23 | **0.419 *** |
| | Sig. (2−tailed) | 0.555 | 0.545 | 0.634 | 0.962 | 0.248 | 0.03 |
| | N | 27 | 27 | 27 | 27 | 27 | 27 |
| **Diesel** | Pearson Correlation | 0.044 | −0.02 | −0.077 | −0.047 | 0.196 | −0.317 |
| | Sig. (2−tailed) | 0.831 | 0.923 | 0.707 | 0.821 | 0.338 | 0.115 |
| | N | 26 | 26 | 26 | 26 | 26 | 26 |
| **LPG** | Spearman's Rho Coefficient | **0.596 **** | −0.225 | −0.01 | 0.285 | 0.28 | **−0.716 **** |
| | Sig. (2−tailed) | 0.001 | 0.259 | 0.96 | 0.15 | 0.158 | <0.001 |
| | N | 27 | 27 | 27 | 27 | 27 | 27 |
| **CNG** | Spearman's Rho Coefficient | −0.203 | 0.321 | 0.063 | −0.332 | **0.517 **** | −0.074 |
| | Sig. (2−tailed) | 0.309 | 0.103 | 0.754 | 0.091 | 0.006 | 0.712 |
| | N | 27 | 27 | 27 | 27 | 27 | 27 |

The next results show that the consumption of LPG in passenger cars is moderately positively linked with the PDI (r = 0.596, sig. = 0.001), and is strongly negatively correlated with IND (r = −0.716, sig. < 0.001). With the increase in the power distance within a society, the consumption of LPG also increases. Nations characterized by restraint will use more LPG-fuelled cars.

The number of cars fuelled by CNG is moderately positively correlated with the tendency of a nation towards long-term orientation (r = 0.517, sig. = 0.006). The results are different compared with other studies, which show that societies with high LTO scores have more pro-environmental behaviours. The results may reflect the perception of the individuals to whom it was promoted as having lower GNG emissions than traditional fuels, when the measurements of the emissions show no difference between them. It can be summarized that individuals with pro-environmental behaviours are willing to buy cars fuelled by CNG, thinking that they are low-emission vehicles.

Correlation can only tell how a variable covaries. To find what cultural dimensions predict the usage of high-emission passenger cars in general, multiple regression analysis should be applied. In order to identify the most significant cultural dimensions that influence the usage of high-emission cars in general, a backward elimination process was

applied. The highest accuracy regression model was obtained for the combination of the UAI and IND independent variables. The results (Table 4) show that the UAI and IND dimensions can describe 42.6% of the variability from the usage of high-emission vehicles in EU-27 (adj. R square = 0.426). At the same time, for 60% of the analysed EU-27 countries, the evaluation error between the predicted values and actual usage percentages of high-emission vehicles is less than 1%. The results of the test were derived from the regression model: $High\_Emiss = 97.44 + 0.03 \cdot UAI - 0.04 \cdot IND$. A one-point increase in the uncertainty avoidance index will increase the usage of high-emission vehicles by 0.03%. Moreover, a one-point increase in the indulgence versus restrain index will decrease the usage of high-emission vehicles by 0.04%.

**Table 4.** Multiple regression analysis between rate of high-emission vehicles and UAI and IND.

| Model Summery | | | |
|---|---|---|---|
| R Square | Adjusted R Square | Std. Error of Estimate | Sig. |
| 0.472 | 0.426 | 1.29636 | <0.001 |
| Predictors: (Constant), IND, UAI; Dependent Variable: *High_Emiss* | | | |
| Parameters | Unstandardized Coefficients ($\beta$) | Std. Error | Sig. |
| (Constant) | 97.441 | 1.281 | <0.001 |
| UAI | 0.032 | 0.012 | 0.017 |
| IND | −0.040 | 0.015 | 0.011 |

Dependent variable: *High_Emiss.*

Figure 11 shows the map of EU countries in relation to the UAI and IND scores. As already mentioned, countries with high UAIs and weak INDs tend to use more high-emission vehicles.

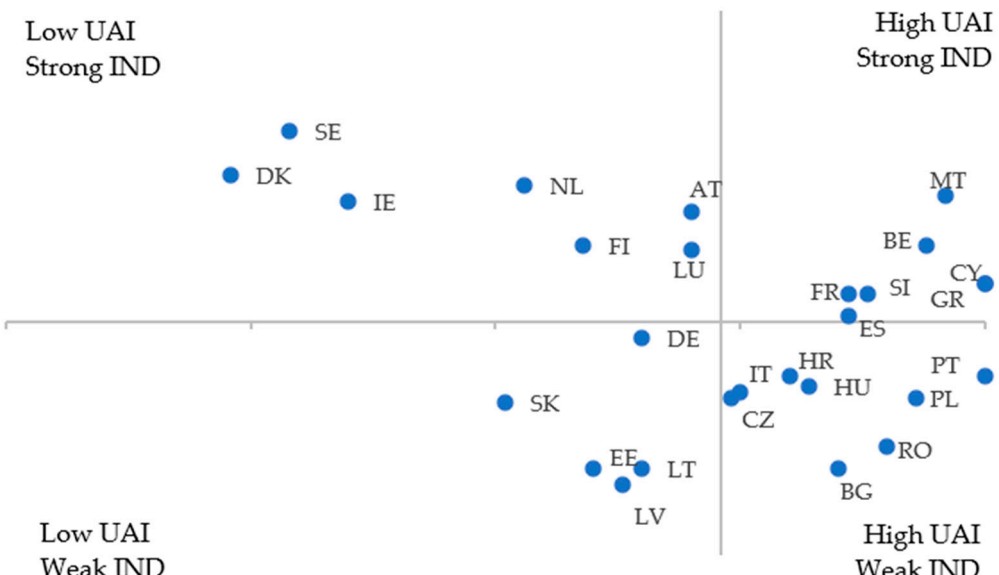

**Figure 11.** EU map, based on uncertainty avoidance index and indulgence versus restraint index. Mean of UAI = 73.03; mean of IND = 42.74.

To increase the usage of medium- and low-emission vehicles in countries such as Bulgaria, Romania, Czechia, Italy, Poland, Portugal, Hungary, and Croatia, the advertising of passenger cars should emphasize the low price for maintenance and the low price of fuels. Moreover, the advertising should be well structured and contain technical information.

### 4.2. Medium-Emission Passenger Cars

Further, the data show that the usage of HEVs and PHEVs (vehicles with medium GNG emissions) is correlated with four out of six national dimensions (Table 5). HEVs are in strong positive correlation with the IND (r = 0.635 > 0.6, sig. < 0.001), in moderate positive correlation with the IDV (r = 0.3 < 0.501< 0.6, sig. = 0.011), in moderate negative correlation with the PDI (r = −0.6 < −0.593 < −0.3, sig. = 0.002), and in moderate negative correlation with the UAI (r = −0.6 < −0.540 < −0.3, sig. = 0.005). PHEVs are in strong positive correlation with the IND (r = 0.703 > 0.6, sig. < 0.001), in moderate positive correlation with the IDV (r = 0.3 < 0.511 < 0.6, sig. = 0.006), in strong negative correlation with the PDI (r = −0.608, sig. = 0.001), and in moderate negative correlation with the UAI (r = −0.6 < −0.407 < −0.3, sig. = 0.035).

**Table 5.** Correlations between cultural dimensions and usage of medium-emission passenger cars. *, **—Automated highlight of SPSS software to underline greatest values.

|  |  | PDI | IDV | MAS | UAI | LTO | IND |
|---|---|---|---|---|---|---|---|
| **HEV** | Pearson Correlation | −0.593 ** | 0.501 * | −0.229 | −0.540 ** | −0.106 | 0.625 ** |
|  | Sig. (2−tailed) | 0.002 | 0.011 | 0.27 | 0.005 | 0.615 | <0.001 |
|  | N | 25 | 25 | 25 | 25 | 25 | 25 |
| **PHEV** | Spearman's Rho Coefficient | −0.608 ** | 0.511 ** | −0.218 | −0.407 * | 0.012 | 0.703 ** |
|  | Sig. (2−tailed) | <0.001 | 0.006 | 0.274 | 0.035 | 0.954 | <0.001 |
|  | N | 27 | 27 | 27 | 27 | 27 | 27 |

The higher power distance index in one country, the less inclined to use hybrid cars are its citizens. The same happens when we analyse the UAI dimension. As theory notes, a high UAI indicates that the individuals in these societies do not easily accept new technologies and innovation. For these individuals, HEVs and PHEVs use a technology that is hard to accept. IDV and IND speak to the existent link between the registered hybrid cars and the national tendency towards individualism and indulgence. People striving for identity and who enjoy life are more inclined to buy hybrid cars.

To investigate the cultural dimensions that can predict the usage of medium-emission vehicles in general, the same methods as previously described were applied. Following the backward elimination process, it was concluded that IDV and IND are the most relevant cultural dimensions. The value of the adjusted R square (0.566) (Table 6) indicates that the variability for hybrid car usage in general can be predicted as 56.6% by the two cultural dimensions: IDV and IND. The multilinear regression model, $Medium\_Emiss = -0.884 + 0.023 \cdot IDV + 0.031 \cdot IND$, can be explained as follows: a one-point increase in the individuality index will increase the usage of medium-emission vehicles by 0.023%.

**Table 6.** Multiple regression analysis between rate of medium-emission vehicles and IDV and IND.

| Model Summery | | | |
|---|---|---|---|
| R | R Square | Adjusted R Square | Sig. |
| 0.755 | 0.600 | 0.566 | <0.001 |
| Predictors: (Constant), IDV, IND; Dependent Variable: *Med_Emiss* | | | |
| Parameters | Unstandardized Coefficients ($\beta$) | Std. Error | Sig. |
| (Constant) | −0.884 | 0.512 | 0.012 |
| IDV | 0.023 | 0.008 | 0.010 |
| IND | 0.031 | 0.007 | <0.001 |

Dependent variable: *Med_Emiss*.

Moreover, a one-point increase in the indulgence versus restraint index will increase the usage of medium-emission vehicles by 0.031%. In countries such as Bulgaria, Romania,

Croatia, Portugal, Cyprus, Greece, and Slovenia, characterized by collectivism and restraint (Figure 12), the advertising of medium-emission vehicles should emphasize the wellbeing of the society when choosing such a vehicle, as well as the low maintenance costs and fuel price.

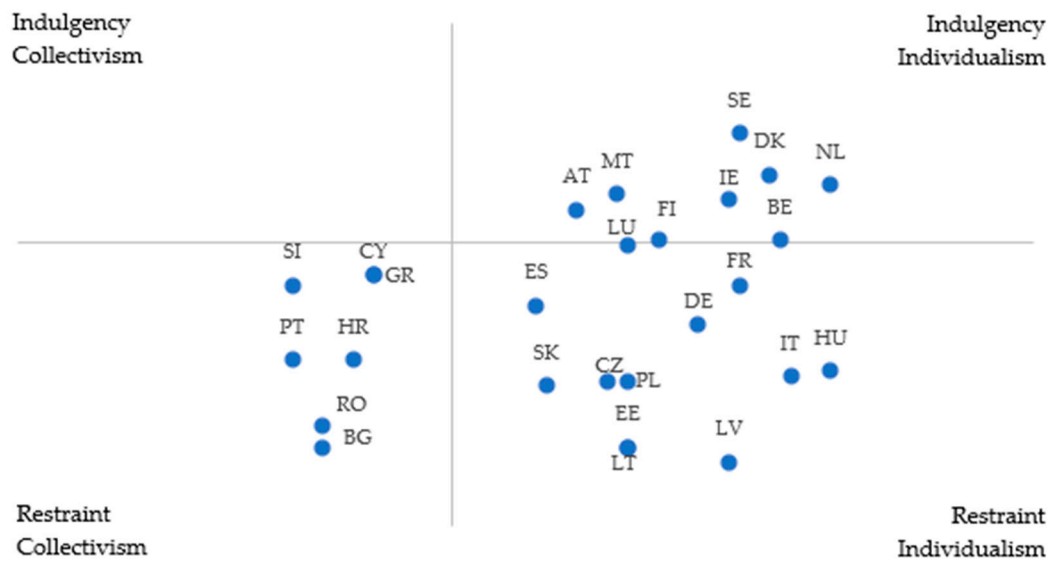

**Figure 12.** EU map, based on individualism versus collectivism and indulgence versus restraint indexes (source). Mean of IND = 42.74; mean of IND = 56.63.

### 4.3. Low-Emission Passenger Cars

The data presented in the next table (Table 7) show similar results to the HEV and PHEV cases. The use of $H_2$ for passenger cars is in moderate negative correlation with the PDI (r = −0.498, sig. = 0.008) and UAI (r = −0.384, sig. = 0.048), but it is in moderate positive correlation with IDV (r = 0.505, sig. = 0.007), and in strong positive correlation with IND (r = 605, sig. < 0.001). The usage of BEVs is in strong negative correlation with the PDI (r = −0.635, sig. = < 0.001), in moderate negative correlation with the UAI (r = −0.425, sig. = 0.027), in moderate positive correlation with IDV (r = 0.443, sig. = 0.021), and in strong positive correlation with IND (r = 639, sig. < 0.001).

**Table 7.** Correlations between cultural dimensions and usage of low-emission passenger cars. *, **—Automated highlight of SPSS software to underline greatest values.

| | | PDI | IDV | MAS | UAI | LTO | IND |
|---|---|---|---|---|---|---|---|
| **H₂** | Spearman's Rho Coefficient | −0.498 ** | 0.505 ** | −0.157 | −0.384 * | 0.211 | 0.605 ** |
| | Sig. (2−tailed) | 0.008 | 0.007 | 0.435 | 0.048 | 0.29 | <0.001 |
| | N | 27 | 27 | 27 | 27 | 27 | 27 |
| **BEV** | Spearman's Rho Coefficient | −0.635 ** | 0.443 * | −0.325 | −0.425 * | 0.003 | 0.639 ** |
| | Sig. (2−tailed) | <0.001 | 0.021 | 0.098 | 0.027 | 0.987 | <0.001 |
| | N | 27 | 27 | 27 | 27 | 27 | 27 |
| **LOW_EMISS** | Spearman's Rho Coefficient | −0.572 ** | 0.425 * | −0.266 | −0.387 * | −0.071 | 0.684 ** |
| | Sig. (2−tailed) | 0.002 | 0.027 | 0.18 | 0.046 | 0.724 | <0.001 |
| | N | 27 | 27 | 27 | 27 | 27 | 27 |

Both low-emission fuels are used mostly by countries dominated by low power distances and low uncertainty avoidance. With increases in the indulgence and individualism cultural dimensions, low-emission fuels are used more often.

As in the previous cases, backward elimination was applied to identify the cultural dimensions that can most accurately predict the usage of low-emission vehicles in general.

It was found (Table 8) that 32.7% of the logarithmic variability in the low-emission passenger car usage can be accounted for by IND (adjusted R square = 0.327). The other cultural dimensions do not have a significant influence on the low-emission vehicle usage variability within EU-27. The regression $Log\_Low\_Emiss = -1.607 + 0.018 \cdot IND$ can be explained by a one-point increase in the indulgence versus restrain index, which will increase the usage of low-emission vehicles by 0.018%.

**Table 8.** Linear regression analysis between rate of low-emission vehicles and IND.

| Model Summery | | | |
|---|---|---|---|
| R | R Square | Adjusted R Square | Sig. |
| 0.594 | 0.352 | 0.327 | <0.001 |
| Predictors: (Constant), IND; Dependent Variable: *Log_Low_Emiss* | | | |
| Parameters | Unstandardized Coefficients ($\beta$) | Std. Error | Sig. |
| (Constant) | −1.607 | 0.222 | 0.011 |
| IND | 0.018 | 0.005 | 0.001 |

Dependent variable: *Log_Low_Emiss.*

## 5. Conclusions

The European Union, a conglomerate of 27 countries with different cultural characteristics, has a common objective to pass from traditional polluting sources to green energy. In the last years, a lot of improvement has been seen, with GHGs dropping in all sectors, with one main exception—transport. Passenger cars and light-duty vehicles are the main pollutants, and together they are responsible for 70% of the total GHG emissions in the EU [11].

The data show that Latvia, Romania, Croatia, Poland, the Czech Republic, Slovakia, Slovenia, Greece, and Italy have the lowest rates of medium- and low-emission passenger vehicles in their car fleets. In order to encourage the usage of the mentioned cars, policy makers have to find more ways to address individuals.

The results of the study show that the consumer preference for one type of fuel when using a passenger car is correlated with at least one of the following four national cultural dimensions: the PDI, IDV, UAI, and IND. With increases in the IDV and IND scores, the usage of low- and medium-emission cars also increases. With increases in the PDI and UAI, the usage of low- and medium-emission cars decreases. The marketing strategies for low- and medium-emission cars should be addressed according to these four cultural factors as well.

Studying the preference for high-emission passenger cars, it can be observed that these are preferred in countries that score high for the uncertainty avoidance index and low for the indulgence versus restrain index. In countries such as Bulgaria, Romania, Czechia, Italy, Poland, Portugal, Hungary, and Croatia, the advertising of medium- and low-emission passenger cars should be well structured and contain technical information, and it should emphasize the low price for maintenance and fuels. Medium- and low-emission passenger cars should be promoted as fashionable items in countries that score high on the PDI.

The driving preference for low- and medium-emission vehicles decreases with the tendency towards collectivism and restraint of EU countries. In Bulgaria, Romania, Croatia, Portugal, Cyprus, Greece, and Slovenia, the advertising of low- and medium-emission vehicles should first create trust and highlight the pro-environmental characteristics, low maintenance costs, and low fuel price.

Our results are in contradiction with other studies that have found a correlation between collectivist cultures and green behaviour. The reasons that could be behind the contradiction will be studied in the future, but the literature suggests that pro-environmental behaviour is linked to the standard of living [66]. However, most of the EU countries characterized by collectivism and restraint are also the countries with the lowest GDPs per

capita [67]. It is understandable that the high price of BEVs, HEVs, PHEVs, and $H_2$ cars is a strong barrier to buying low-emission cars.

The symbolism of BEVs, HEVs, and PHEVs must also be considered. For many, it is more important to be "seen green" than "to be green" [68]. Further, we can argue that wanting to be seen as pro-environmental, and different from others, shows the tendency towards individuality rather than collectivist behaviour. Our study, which uses official data on the passenger cars that are already registered (2019), does not consider the attitudes and perceptions towards low- and medium-emission cars, but the actions taken by EU citizens.

As a potential area for future research in this study area, the authors intend to conduct a similar analysis for the pandemic and the period of conflict in Ukraine, which would be highly valuable. This would enable an examination of the changes that have arisen as a result of these exceptional circumstances, and their impacts on the findings of the current article.

**Author Contributions:** Conceptualization, I.A.I., D.S. and S.D.C.; Methodology, I.A.I., P.H., D.D.M. and S.D.C.; Software, L.C.; Validation, D.D.M., L.C. and S.D.C.; Formal analysis, P.H.; Investigation, I.A.I., P.H., D.D.M., D.S. and L.C.; Resources, I.A.I.; Data curation, P.H. and L.C.; Writing—original draft, D.D.M., D.S. and S.D.C.; Writing—review & editing, D.S. and S.D.C.; Funding acquisition, I.A.I. All authors have read and agreed to the published version of the manuscript.

**Funding:** This research and the APC were funded by European Union's Horizon 2020 research and innovation programme under the Marie Sklodowska-Curie grant 801505.

**Institutional Review Board Statement:** Not applicable.

**Conflicts of Interest:** The authors declare no conflict of interest. The funders had no role in the design of the study; in the collection, analyses, or interpretation of data; in the writing of the manuscript; or in the decision to publish the results.

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
