# Peer review of "The Influence of Cultural Factors on Choosing Low-Emission Passenger Cars"

_sustainability, doi:10.3390/su15086848_

Round 1

Reviewer 1 Report

Dear Authors, thank you for this timely and intersting work related to behavioural aspects and Hofstede's model for car passenger in fleets. Moreover, the impact of this research is wide, as it takes into consideration many countries, spanning across the EU.

Therefore, the generalizability of this research is very high, as well as its interest to the readership. The methodology is clearly applied and presented. Also the jargon utilized is compliant with the readership of the research field.

However, as a minor suggestion, please note that some further results worth mentioning in literature section on the general mobility literature: (1) https://link.springer.com/chapter/10.1007/978-3-319-29504-6_21 ; (2) https://www.mdpi.com/2076-3417/11/4/1578 .

Author Response

Submitted manuscript ID: sustainability-2300691

Dear reviewers,

 We express our gratitude for the chance to submit the revised version of our manuscript to the Sustainability Journal. The authors sincerely appreciate the valuable feedback and suggestions provided by the Editors and Reviewers, which have undoubtedly enhanced the manuscript's quality. We have devoted considerable effort to addressing your feedback, and we trust that this updated version will fulfill your expectations.

In the following, author answers and changes to the manuscript are presented.

Please note that major modifications are highlighted in red in the revised manuscript.

Reviewer 2 Report

Authors have developed rather interesting research on an urgent topic. The paper is relatively well structured and based on comprehensive analytical data. Nevertheless, there are some recommendations that might improve the quality of the paper:

1) Section 1 introduces the research problem's general urgency and formulation of the research objective. Nonetheless, the paper structure might be more transparent if authors clarify the research goal and subtasks in a more precise and detailed way;

2) The authors provide essential analytical information, but some data do not have time markers. It is recommended to clarify a period for which this data is relevant (for example, in lines 39-41, there is no information for the period for which data is presented, while in line 58, it is mentioned);

3) literature review is well-organized and informative, but there is a lack of argumentation for choosing as the best proxies of the cultural profile six dimensions of Hofstede's Model. It seems that there is a necessity for more in-depth argumentation, especially in comparison with other cultural determinants used by scientists in earlier research. Moreover, if authors declare using of "backwards elimination method" to develop the best regression model for the specific dependent variable, maybe it can be more valuable to expand a set of explanatory variables for the preliminary research stage;

4) It is necessary to provide a detailed description of cultural dimensions used in the paper in the research methodology or results section. For example, the statement "The results (Table 4) show that UAI and IND dimensions can predict 42.6% from the usage of high emission vehicles, in EU-27 (Adj. R Square = 0.426)" does not bring to a clear understanding of the research results for non-experienced readers;

5) Research results. Table 3 demonstrates that correlation coefficients are relatively high for such dimensions as IND, LTO and PDI, but for the regression model selected UAI and IND. Table 5 indicates that PDI, IDV, UAI and IND correlate highly with dependent variables, but the regression model consists of only IDV and IND. Table 7 also demonstrates that PDI, IDV, UAI and IND correlate highly with dependent variables, but regression results demonstrate only IND as an explanatory variable. I assume that authors include only statistically significant results to Tables with regression modelling results, but such a selective approach leads to misunderstanding. There is necessary to provide full regression results or (if my assumption is wrong) to use all those variables that demonstrated comparably high correlation with dependent variables in the regression. Moreover, it is better to expand the period of observation to obtain reliable modelling results because modelling based only on one-year observation does not allow to get comprehensive results and their progress;

6) Conclusions should include a comparison of the obtained results with the existing ones.

Author Response

Dear Reviewers, 

 We express our gratitude for the chance to submit the revised version of our manuscript to the Sustainability Journal. The authors sincerely appreciate the valuable feedback and suggestions provided by the Editors and Reviewers, which have undoubtedly enhanced the manuscript's quality. We have devoted considerable effort to addressing your feedback, and we trust that this updated version will fulfill your expectations. 

In the following, author answers and changes to the manuscript are presented. 

Reviewer 3 Report

The paper needs improvements as follows:

Title: As it does not reflect the paper's content – We did not find enough details about "the relations between national cultural dimensions… in the EU". The content of subsection 2.2. "Cultural factors influencing buying behaviour of passenger cars" does not present the abovementioned relations. Details about the improvement of section 2, including subsection 2.2. will be proved further.

Abstract: The problem statement and the aim of the study should be clearly stated. The abstract, as is, does not provide a concise account of the work and conclusion of the research study. It needs to be more structured and synthesized for research clarity. The manuscript lacks a clearly structured research methodology.

Keywords: We recommend avoiding using abbreviations in this section

Introduction: We did not find clearly presented the objectives or the research questions. The authors should

elaborate on relevant contextual ideas and the background leading to research studies, and explain why it is essential for this research study.

We recommend focusing on the research framework to avoid overstating your research scope.

Literature review: The critical dialog with literature is missing. Subsection 2.2. does not clearly reflect its content.

We expected to find a detailed presentation of Hofstede's six cultural dimensions for the EU-27.

Why are the data from 2019 (before the COVID-19 crisis), and we are in 2023 (after the COVID-19 crisis)?

The following figures should be redrawn using new data:

Figure 1. Average NECD Emissions (gCOâ‚‚ /km) from new passenger cars in EU countries (2020)

Figure 2. Percent of passenger cars in EU-27 (except Bulgaria), with high COâ‚‚ emissions, in 2019

Figure 3. Percent of passenger cars in EU-27 (except Bulgaria), with medium COâ‚‚ emission, in 2019

Figure 4. Percent of passenger cars in EU-27, with low COâ‚‚ emission, in 2019

Research methodology: It is the main weakness of the paper and raises many questions. It is unclear how the authors calculated Hofstede's six cultural dimensions for the EU-27.

We recommend the authors seriously improve this section by adding more details regarding the research data collection and analysis.

It is compulsory to briefly describe the methods of data employed and their application and appropriateness for data analysis.

Results and Discussions:

The results were not well-presented to readers to understand the focus of the research study.

Tables are not critically analyzed, which can provide an unclear path for future researchers to replicate the study.

The results must be interpretive rather than just descriptive and connect the research results with relevant literature citations for validity and reliability.

The discussion does not integrate with the results of the research study. Discussing the results could be improved by interpreting them to support theories related to the research topic.

There was no discussion of the cultural factors and under which circumstances these factors were conducive to or inhibited the behavior of passenger cars.

We have some doubts about the accuracy of Figure 5. EU map, based on Uncertainty Avoidance Index and Indulgence versus Restrain Index and Figure 6. The EU map is based on the Individualism versus Collectivism and Indulgence versus Restraint Index (source).

Conclusions: The conclusions are not supported by the research data, which does not indicate a clearer path for future studies on the topic.

A follow-up of restated results with supporting literature reviews could make the conclusion section more effective.

Good luck!

Author Response

(The authors gave the same response as above.)

Round 2

Reviewer 2 Report

All the recommendations concerning the improvement of the paper are considered (or explained) be the authors. There are no further recommendations.

Best regards

Reviewer 3 Report

Good luck!